# Volatile Profile of Dry and Wet Aged Beef Loin and Its Relationship with Consumer Flavour Liking

**DOI:** 10.3390/foods10123113

**Published:** 2021-12-15

**Authors:** Zhenzhao Li, Minh Ha, Damian Frank, Peter McGilchrist, Robyn Dorothy Warner

**Affiliations:** 1Faculty of Veterinary and Agricultural Sciences, School of Agriculture and Food, The University of Melbourne, Parkville, VIC 3010, Australia; zhenzhaol@student.unimelb.edu.au (Z.L.); robyn.warner@unimelb.edu.au (R.D.W.); 2Commonwealth Scientific and Industry Research Organisation, 11 Julius Ave, North Ryde, NSW 2113, Australia; damian.frank@sydney.edu.au; 3School of Environmental and Rural Science, University of New England, Armidale, NSW 2351, Australia; pmcgilc2@une.edu.au

**Keywords:** Headspace-Solid-Phase Microextraction Gas Chromatography-Mass Spectrometry (HS-SPME GC-MS), Gas Chromatography-Olfactometry-Mass Spectrometry (GC-O-MS), meat ageing, odour active, beef flavour

## Abstract

This study investigated the effect of ageing method and ageing time on the volatile profiles of grilled beef striploins (*Longissimus thoracis et lumborum*) and their relationship with consumer flavour liking. Volatiles were measured in grilled steaks subjected to 35 days of dry ageing, 35 days of wet ageing, 56 days of dry ageing or 56 days of wet ageing, using headspace-solid-phase microextraction followed by gas chromatography-mass spectrometry. Gas chromatography-olfactometry-mass spectrometry was also conducted on 35-day wet and dry aged samples to identify volatiles with high odour impact. The concentration of many odour impact volatiles, e.g., 3-hydroxy-2-butanone, 2-acetyl-2-thiazoline, and various alkyl-pyrazines, was significantly higher in dry aged beef compared to wet aged beef (*p* < 0.05). Several odour impact volatiles, e.g., 2-acetyl-1-pyrroline, and alkyl-pyrazines, decreased significantly with ageing time (*p* < 0.05), while volatile products of lipid oxidation and microbial metabolism increased with ageing time. Partial least-squares regression analysis showed that the higher consumer flavour liking for 35-day dry aged beef was associated with higher concentrations of desirable odour-active volatiles.

## 1. Introduction

Ageing in vacuum packs is widely used in the beef industry to preserve and to improve the eating quality of meat [1]. Ageing of beef is most commonly achieved by the storage of primals or steaks in vacuum packaging under controlled low temperature to improve the tenderness and extend the shelf life [2]. This process is usually termed ‘wet’ ageing. In contrast, ‘dry’ ageing of beef refers to the ageing of unpackaged primals in air under strictly controlled conditions including temperature, moisture, and air velocity [3]. Dry aged beef is often marketed as a premium product with improved flavour by high-end butchers and restaurants, although it remains controversial whether the sensory quality of dry aged beef is higher than that of wet aged beef [4]. A more recent study showed that dry aged beef received significantly higher overall and flavour liking scores compared to wet aged counterparts using Meat Standard Australia (MSA) consumer panels with 1440 consumers [3]. Although both ageing methods improve most palatability attributes of beef, wet aged meat was reported to be associated with negative flavours including sour, ‘serumy’, and metallic, whereas the dry ageing is known to enhance the positive flavours, such as nutty, roasted, and buttery, in beef [5].

Due to the consensus among consumers and retailers that dry ageing significantly improves the flavour of beef compared to that of wet ageing, the flavour chemistry of dry aged beef has been a subject of investigation in several studies [4,6]. Iida, et al. [6] showed that the difference in flavour between dry and wet aged beef could be partially attributed to the increase in umami taste of dry aged beef. While most studies of dry/wet ageing of beef were focused on the sensory assessment and the taste-active compounds, the role of flavour-active volatiles in the flavour of aged beef is not well understood. Volatiles formed in cooked beef have been proven to play an essential role in the perception of flavour [7,8]. The characteristic aroma of cooked beef is largely due to volatile substances formed during cooking [9]. King, et al. [10] and Utama, et al. [11] reported that the volatile profile of dry aged beef is different from that of wet aged beef. However, the relationship between sensory flavour liking, and volatiles was not explored in their research. Aside from the ageing method (dry or wet), the period of ageing is another determinant of the palatability of aged beef. Dry ageing of beef for more than 40 days is reported to negatively impact on eating quality due to increased lipid oxidation and microbial spoilage [3,6]. The influence of ageing time on volatiles in cooked, wet aged beef was shown in the study of Frank, et al. [12] and Watanabe, et al. [13]. While the change of volatiles in dry ageing of beef and its comparison with wet ageing are rarely reported.

The primary goal of this study was to analyse the effects of dry and wet ageing, and ageing time, on volatile profiles of beef. The relationship between volatiles and flavour liking of aged beef was also investigated to explain the difference in flavour liking of beef aged with different ageing methods.

## 2. Materials and Methods

### 2.1. Animal and Carcass Collection

The sample collection process for this project was described fully in the study of Ha, et al. [3]. The cattle were mixed breeds of predominantly Angus, Hereford and Murray Grey, <24 months old and were hormone- and antibiotic-free. Briefly, carcasses (*n* = 24) from 16 steers and eight heifers with normal pH (<5.7) were selected at 24 h post mortem from a commercial beef processing plant in Tasmania, Australia. Full length bone-in *longissimus thoracis et lumborum* was excised from both sides of 24 carcasses. All primals were vacuum packed and transported in a refrigerated lorry to Top Cut Foods (Gold Coast, QLD) for further processing and ageing.

### 2.2. Ageing Specification

The wet and dry ageing conditions were described in the study of Ha, et al. [3]. Upon arriving at Top Cut Foods, primals were boned into striploins (boneless) or OP (Oven Prepared) ribs (bone-in). The ageing method by ageing combinations were allocated within two *longissimus thoracis et lumborum* primals from one carcass. There were seven combinations of treatments, and only the 35-day wet and dry aged, and the 56-day wet and dry aged samples from 12 randomly selected carcasses were used in the present study. Wet ageing was conducted with boneless primals in Cryovac^®^ polyamide, polyethylene vacuum pack bags with an oxygen transmission rate of 20 cc/m^2^/24 h at 23 °C without illumination. The primals were placed in a refrigerated room with temperature fluctuating between 2–6 °C.

The dry ageing room was a multi-batch chiller with movable racks and two UV lights fitted to its ceiling. The meat samples were rotated daily to different positions in the chiller. The relative humidity (RH) in the chamber ranged from 53% to 100% with an average RH of 89.4% over the experimental period. The recorded temperature was 1.3 °C to 4.2 °C with an average temperature of 2.1 °C. The airspeed at the central position varied between 0.75 and 1.2 m/s.

After each ageing period, steaks 2.5 cm thick, and small samples, were obtained from the primals and frozen at −20 °C for 3 months until sensory analysis or at −80 °C for four months until flavour chemistry analysis. Consumer flavour liking was obtained using Meat Standards Australia (MSA) untrained consumer panels with 900 participants as described by Ha, et al. [3].

### 2.3. Intramuscular Fat Analysis

Intramuscular fat (IMF) was measured using near-infrared spectrometry as described by Perry, et al. [14]. Frozen samples (approximately 100 g) were freeze-dried and finely ground. The ground samples were analysed with a Technicon InfrAlyser 450 spectrometer (Bran and Luebbe, Sydney, Australia) and expressed as percentage of IMF in raw meat (*w/w*).

### 2.4. Headspace-Solid-Phase Microextraction Gas Chromatography-Mass Spectrometry (HS-SPME GC-MS)

Beef steaks were thawed overnight at 4 °C before testing and grilled for 3 min using a clamshell grill (Silex, Marrickville, Australia) set at 220 °C. The samples were then allowed to rest for another 3 min under aluminium foil. The grilled beef steaks were then weighed, roughly cut and Milli-Q water was added at a ratio of 1:2 (Milli-Q water:meat). Samples were macerated using a hand-held food processor and 4 g of slurry was transferred to a headspace vial. The internal standard 4-methyl-1-pentanol was placed into a 200 µL insert. Duplicate samples were placed in the autosampler (AOC-5000, Shimadzu, Rydalmere, Australia). The samples were pre-incubated at 40 °C for 15 min and the headspace volatiles were extracted with divinylbenzene/carboxen/PDMS 23-gauge, 2 cm solid phase microextraction (SPME) fibres (Supelco, Sigma-Aldrich, Castle Hill, Australia) for 40 min at 40 °C with agitation. The extracted volatiles were desorbed in splitless mode into a hot injector (250 °C) for 5 min and separated using gas chromatography–mass spectrometry (QP-2010-Plus GC-MS, Shimadzu, Rydalmere, Australia) on a Zebron- Wax column (Phenomenex, Lane Cove West, Australia, 30 m, 0.25 id, 0.25 μm film) with the following temperature programming; initial temperature of 35 °C was held for 5 min and then heated at 5 °C/min to 250 °C. The electron impact (EI) mass spectrometer was programmed to scan the mass range *m*/*z* 40–250. An aliphatic hydrocarbon mix (C8-C32, Sigma-Aldrich, Castle Hill, Australia) was used to determine linear retention indices. Compounds were identified by comparing their electron impact mass spectra with reference spectra in the National Institute of Standards and Technology mass spectral database (NIST 2002) and by linear retention indices matching those of published values (NIST Chemistry WebBook). Integrated area data were normalised to the internal standard and semi-quantitative data (µg/g) were estimated.

### 2.5. Gas Chromatography-Olfactometry-Mass Spectrometry (GC-O-MS)

GC-O-MS analysis was conducted on a subset of 35-day dry aged and wet aged samples (*n* = 8 carcasses). A panel of trained GC-O assessors (*n* = 6) evaluated the column effluent of each matching pair (from the same carcass) of wet and dry aged samples using a validated direct intensity technique. Assessors were trained according to previously reported protocols [15]. Volatiles were trapped onto Tenax traps and desorbed using a short path thermal desorption unit (Scientific Services, Ringoes, NJ, USA) onto the injector port of the GC-O-MS and separated on a Zebron-Wax capillary column as described previously [8]. Briefly, assessors measured the odour intensity of the GC effluent using a computer mouse and time intensity software SensoMaker [16]. The odour intensity throughout the chromatographic run (approximately 20 min) was rated continuously using an unstructured 10 cm line scale on a computer screen, where 0 represented the absence of any perceived odour, 2.5 was used to indicate an odour of mild intensity, 5.0 moderate intensity, 7.5 strong and 10 very strong. Odour intensity data were continuously acquired at a rate of 1 Hz. In case of odours persisting for several seconds, assessors were asked to continuously rate the intensity until the odour stimulus disappeared. Simultaneously, assessors were asked to verbally describe the odour quality using a microphone. Assessor descriptions were digitally recorded (GoldWave Inc., St John’s, NL, Canada). Time intensity data from each panellist were imported into Microsoft Excel and annotated with odour descriptors (when given) and matched to specific volatiles based on compounds identified eluting at the same time; for example, from the electron impact mass spectral data in National Institute of Standards and Technology library. For each distinct odour event, the integrated area under the time curve (AUC) was calculated, for example intensity (1–10) × duration (seconds). Replicate AUC data were used for statistical analysis and the average AUC was used to construct aromagrams. Peaks detected by less than two assessors were considered noise and deleted from the aromagram. As there was no time delay between the GC-MS and the olfactory port effluent, odours and volatiles could be accurately matched. Retention indices were calculated for volatiles on the GC-O to enable cross-referencing to the SPME data.

### 2.6. Statistical Analysis

Volatile data were analysed using GenStat^®^ (16th Edition, VSN International, Hemel Hempstead, UK). The multivariate analysis of variance (MANOVA) was performed on the volatile data using the following model; Response (volatile data) = ageing method (main effect) + ageing time (main effect) + ageing method ∗ ageing time (interaction) + animal (block) + intramuscular fat content (covariate). The average standard error of difference (SED) were calculated for main effects and interactions. Effects of treatments were considered significant if the difference between treatments were greater than 2 × SED.

The consumer flavour liking scores reported in Ha, et al. [3] were matched with mean integrated volatile data from samples from the same carcass and treatment group (*n* =12 carcasses in each ageing method ∗ ageing time treatment group, there was one carcass with missing sensory data in 56-day dry ageing and one in 56-day wet ageing groups respectively). Principal component analysis (PCA) was performed on flavour liking data of samples and their corresponding semi-quantitative volatile data with confirmed odour impact in GC-O analysis or literatures using Matlab (R2019b, The MathWorks Inc., Natick, MA, United States). A measure of sample adequacy was also performed using the Kaiser–Meyer–Olkin (KMO) test to check the suitability of the selected variables for PCA. A partial least-squares regression analysis was performed on the mean-integrated volatile and sensory data for each treatment group using the partial least-squares (PLS) procedure in Genstat following the method described by Frank, et al. [15]. All data were first standardised using the z-score function in Matlab. All volatiles results were used in the initial model, and the X-component loadings were used to select the most influential volatiles for flavour liking of aged beef. A subset of volatiles (*n* = 22) known to be odour-active in the GC-O results or the literature were selected for the final optimised PLS model. One latent factor was used in modelling with cross-validation procedures (4 groups, random seeds = 2). The Osten’s *F*-test was performed on the predicted sum of square statistic to evaluate the significance of latent factor. The root mean square error of cross validation was calculated from the predicted residual sum of square statistic.

## 3. Results and Discussion

### 3.1. Grilled Beef Volatile Profiles

A total of 62 volatiles were identified in the grilled beef headspace extracts based on EI mass spectra and retention indices (Table 1). 2-Methylbutanal, ethanol, and 3-hydroxy-2-butanone were quantitatively the most abundant volatiles. Hexanal, butyl formate, 3-methylbutanal, 2,6-dimethylpyrazine, 2-pentanone, trimethylpyrazine, 2,5-dimethylbenzaldehyde, and 2-ethyl-3,5 dimethylpyrazine were also measured at relatively high concentrations. Most of these volatiles have been reported in beef aroma extracts previously [8,17,18]. Among these compounds, 3-hydroxy-2-butanone was shown to be associated with aged meat [18]. Significant differences were measured for several odour-active volatiles for the main effects of “ageing method” or “ageing time” and their interaction (Table 1). Overall, the dry aged and wet aged beef had similar intensity of total volatiles. A significantly higher concentration of alcohols, especially ethanol, was measured in the wet aged beef compared to dry aged counterparts, and the concentration of ethanol increased significantly over time in wet ageing. The increased ethanol in wet aged beef is likely to be a result of lactic acid bacteria fermentation as reported previously in vacuum-packed beef [12,19,20]. Similarly, the concentration of acetic acid was significantly higher in wet aged beef. The finding is consistent with previous studies and the higher acetate could be explained by the fermentation of microflora [20]. It is worth noting that many other organic acids might also increase during vacuum storage, but they are not volatile and thus not measured easily by HS-SPME GC-MS. Esters are known to be formed in the esterification process in the cooking of beef or by microbial spoilage [21]. In the present study, butyl formate was detected at a relatively high concentration in both dry aged and wet aged beef but decreased with the ageing time. Butyl formate is rarely reported in the headspace of grill beef, and a previous study indicates that it could be formed in an unusual microbial metabolism in the wet ageing of beef [22]. Ketones such as 3-hydroxy-2-butanone and 2-pentanone were significantly different between ageing methods. The concentration of 3-hydroxy-2-butanone was significantly higher in dry aged beef than that in wet aged beef. The 3-hydroxy-2-butanone is known as an oxidative product of saturated fat but has also been shown to be related to the Maillard reaction [23]. 3-Hydroxy-2-butanone is considered an important contributor to the buttery odour in beef [24]. The concentration of acetone increased significantly over time regardless of the ageing method. The increase of acetone in the headspace of meat was also reported to be associated with microbial metabolism during storage [25].

Aldehydes generally play important roles in the flavour of beef and quantitatively dominate many other odour-active volatiles in cooked beef [9]. In meat, aldehydes are mainly formed during lipid oxidation (e.g., hexanal, octanal, nonanal) and the Strecker degradation of amino acids, such as 2-methylbutanal, 3-methylbutanal and benzaldehyde [26]. In the present study, the concentration of 2-methylbutanal and 3-methylbutanal did not differ with ageing method or ageing time, but the concentration of 2,5-dimethylbenzaldehyde decreased significantly with ageing time. The Strecker degradation of amino acids is primarily caused by its reaction with dicarbonyl compounds formed in the Maillard reaction [9]. In the ageing of beef, the concentration of free amino acids has been reported to increase continuously due to proteolysis [12,27]. Therefore, we speculate that the lower extent of Strecker degradation of 56-day aged beef was likely due to lower availability of dicarbonyl compounds from the Maillard reaction [28,29]. This hypothesis is further supported by the lower extent of Maillard reaction in 56-day aged beef, as indicated by the lower concentrations of alkyl-pyrazines, compared to its 35-day aged counterpart measured in the present study. Hexanal, heptanal, octanal, and nonanal, are commonly reported lipid-derived volatiles in cooked beef, and their odours are described as ‘green’, ‘fatty’, and ‘sweet’ [30] and are commonly used as the indicators of lipid oxidation in meat [31]. In our study, the concentrations of heptanal, octanal, and nonanal increased with the ageing time in dry ageing (*p* < 0.05 except for heptanal) but decreased with the ageing in wet ageing (*p* < 0.05 for all). In the beef aged 56 days, the concentrations of these volatiles were higher in the dry aged meat compared to the wet aged meat (*p* < 0.05). The increase of these volatiles in dry ageing agrees with the results of thiobarbituric acid reactive substances (TBARS) obtained in the same samples [3]. However, the TBARS in the wet aged beef increased with ageing time in the study of Ha, et al. [3], which is contradictory to the volatile results. Therefore, the relationship between volatile compounds and TBARS was not clear in the present study, and a possible explanation for this is the conversion of volatiles aldehydes into organic acids due to the microbial activities in the wet ageing [32]. Whereas the malondialdehyde measured by TBARS is unlikely to be affected by the microorganisms in meat during the wet ageing.

Significant effects of ageing method and ageing time were measured for alkyl-pyrazine compounds in the present study (Table 1). Specifically, 2,5-dimethylpyrazine, 2,6-dimethylpyrazine, 2-ethyl-5-methylpyrazine, 2-ethyl-6-methylpyrazine, trimethyl pyrazine, 2-ethyl-3,5-dimethylpyrazine, and 3-ethyl-2,5-dimethylpyrazine were detected at relatively high abundance. The concentrations of most of these compounds were significantly higher in the dry aged beef compared to wet aged samples (*p* < 0.05, except for 2,5-dimethylpyrazine and 2,6-dimethylpyrazine), and the concentration tended to decrease with ageing time in both ageing methods (*p* < 0.05, except for 2,6-dimethylpyrazine and trimethylpyrazine). This result indicates a higher rate of Maillard reaction in dry aged beef compared to wet aged. The difference in Maillard reaction could be attributed to the ultimate pH as reported by Ha, et al. [3]. The dry aged beef had significantly higher pH compared to that of wet aged beef after ageing (*p* < 0.001), and the pH declined in both ageing methods with the ageing time. The study of Madruga and Mottram [33] shows that the formation of alkyl-pyrazines during the cooking of meat is pH dependent, and a higher pH contributed to its generation. It was postulated in their studies that the unprotonated amino acids are higher in high pH and thus favour their condensation with reducing sugars. Also, the higher alkyl-pyrazines in the dry aged beef could be related to the increase of lipid oxidation in it during ageing, as Ha, et al. [3] showed that the dry aged beef had significantly higher TBARS than the wet aged beef at day 56 (*p* < 0.05). In addition to the Maillard reaction, the alkyl-pyrazines are known to be formed in the lipid-Maillard interaction [9,29]. The aldehydes formed in lipid oxidation could compete with the dicarbonyl formed from sugar for the amino acids to form alkyl-pyrazines [29]. Alkyl- pyrazines are well known to impart meaty and roasty flavour in cooked beef [8,9]. Therefore, the effect of ageing on pH or lipid oxidation could be translated to difference in volatiles compounds including pyrazines and subsequently alter the flavour attribute of beef. The higher pH of dry aged beef has been reported in other studies [34,35]. Aside from alkyl-pyrazines, ageing time or ageing method significantly impacted on many other Maillard-derived volatiles such as pyridines and pyrroles although these compounds were detected at relatively low concentrations.

### 3.2. Gas Chromatography-Olfactometry

The average aromagram for the wet and dry aged grilled beef samples at 35-days ageing are shown in Figure 1 as well as the trained panel description for key peaks, and odour intensity of volatile compounds eluted at different retention times (Figure 1). Although for most aroma peaks, only small differences were measured between ageing methods, the odour intensity of several compounds were higher in the dry aged samples. Specifically, the aromagram of dry aged beef is characterised by higher earthy (acetone, 2-methylpropnanal), meaty (3,5-diethyl-2-methyl-pyrazine), barbecue ((*E*)-2-nonenal), roasted (2-acetyl-2-thiazoline), and fatty (*p*-cresol) odours. The main grilled beef aroma peaks correspond with the Maillard-derived alkyl-pyrazines, which are well-known components of beef aroma [36]. The aromagram was similar to that previously reported in beef [8]. The higher intensity of these odour-active volatiles in dry aged beef supports the higher flavour liking score in the sensory assessment using associated samples [3]. No off-flavours were detected in the aroma profiles of both ageing methods.

### 3.3. Relationship between Flavour Liking and Volatiles of Aged Beef

Multivariate analysis was conducted on the GC-MS volatiles and consumer flavour liking data previously reported in Ha, et al. [3] to investigate the association between volatiles and consumer flavour liking of aged beef. Volatile profiles of aged beef and their flavour liking scores are summarised by the PCA biplot (Figure 2). The result of the KMO test indicated that the selected variables are appropriate for PCA (KMO = 0.67). Although there was variance between carcasses within each treatment group, the effects of ageing method and ageing time on volatile profile of beef are clearly presented in the biplot. Beef aged for different periods are separated along PC1 and PC2. With increased ageing time, the concentrations of most odour impact volatiles decreased. The products of Maillard reaction and Strecker degradation (eg.2-acetyl-1-pyroline and alkyl-pyrazines) decreased significantly with ageing time after 35 days (*p* < 0.05), whereas a small group of lipid oxidation products (heptanal, decanal) and ethanol increased (*p* < 0.05). The study of Watanabe, et al. [13] showed that most volatiles derived from Maillard, Strecker degradation, and lipid oxidation increased significantly during the wet ageing of beef *M. biceps femoris* for 2–30 days, indicating that the effect of ageing time on cooked beef volatiles could differ between cuts. A study on dry aged Hanwoo *longissimus thoracis et lumborum* showed a significant increase of volatile concentrations from day 0 to day 40, however, the change of volatile concentration after day 40 was not reported [11]. In the present study, the decline in volatile concentration with increased ageing time (35 vs. 56 days) is reflected by the decrease of flavour liking scores over the same time period in both dry and wet aged beef samples [3]. PC1 andPC2 also distinguished the volatile profiles of dry aged beef from its wet aged counterpart. The dry aged beef generally contained higher concentrations of odour impact volatiles compared to that of wet aged beef. This finding agrees with previous studies in beef ageing. King, et al. [10] reported that 14-day dry aged beef exhibited higher concentrations of most volatiles than that in wet aged beef, except for total aldehydes and ketones. Among the four combinations of ageing methods and time (wet aged 35 days and 56 days; and dry aged 35 days and 56 days), the 35-day dry aged beef was characterised by the highest concentration of many odour impact volatiles including 2-ethyl-6-methylpyrazine, 2-ethyl-5-methyl pyrazine, 3-hydroxy-2-butanone, 2-acetyl-2-thiazoline and trimethyl pyrazine, explaining the high flavour liking scores in the consumer sensory assessment [3]. According to Kilgannon, et al. [37], the concentration of many of these volatiles are positively correlated to the flavour liking of wet aged beef measured using an MSA consumer sensory panel. Therefore, 35-day dry ageing led to a volatile profile with an increase in desirable volatiles preferred by consumers.

To better understand the effect of volatiles on flavour liking of dry/wet aged beef, a PLS regression model was established using flavour liking of aged beef samples as Y and their corresponding concentration of selected volatiles as X (Table 2). A moderate PLS model was obtained using only one latent factor. Similar to the outcome of PCA, flavour liking was positively correlated to most volatiles. A positive relationship was measured between flavour liking and the concentration of alkyl-pyrazines, which agrees with the finding of Frank, et al. [15]. Frank, et al. [15] reported that 2-ethyl-3,5-dimethylpyrazine was an important contributor to the grilled flavour and flavour impact attributes in lamb, while the trimethylpyrazine was positively related to the aged meat flavour. Alkyl-pyrazines are commonly reported to impart desirable flavour attributes such as chocolate and roasted in cooked meat [8,15,38]. Similarly, 3-hydroxy-2-butanone, 2-acetyl-1-pyrroline, and 2-acetyl-2-thiazoline have been reported to positively influence the flavour profile of cooked meat [15,26]. Hence, it is postulated that the higher concentration of alkyl-pyrazines, 2-acetyl-2-thiazoline, and 3-hydroxy-2-butanone, in dry aged beef contributed to its high flavour liking in sensory assessment. Additionally, the decline in alkyl-pyrazines and 2-acetyl-1-pyrroline with the ageing time could be responsible for the decrease of flavour liking with time of ageing. The flavour liking was negatively correlated to the concentrations of acetic acid and ethanol, which accumulated in the wet ageing process most likely due to microbial fermentation in the present study. Flavour liking score was also positively correlated to several aldehydes produced in lipid oxidation including hexanal, 2-methylbutanal, and nonanal. This result is similar to those reported by Frank, et al. [15] and Song, et al. [38]. The study of Song, et al. [39] showed that a moderate lipid oxidation is essential for the formation of beef flavour in cooking. However, in the 56-day dry aged beef, the concentration of heptanal and octanal reached a relatively high level, which may have caused a perception of oxidised flavour by consumers and thus reduced the sensory score as reported previously studies [7]. The concentration of 3-hydroxy-2-butanone was positively related to the flavour liking in our study. A similar finding was reported by Legako, et al. [26]. However, the study of Stetzer, et al. [18] showed that the 3-hydroxy-2-butanone was correlated to the ‘livery’ off-flavour in aged beef. In summary, the PLSR model further supports that the volatile profile was an important driver of the flavour liking of aged beef. However, the model should be interpreted cautiously as it was based on a relatively small set of observations (*n* = 4, mean-centred values for treatment combinations) and does not take into consideration the important role of non-volatile flavour compounds such as free amino acids, peptides and nucleotides. In addition, the roles of particular volatiles in flavour liking needs to be further confirmed in omission and addition tests as suggested by Frank, et al. [15].

## 4. Conclusions

Results from this study indicated that the dry aged and wet aged beef exhibited similar volatile profiles measured by GC-MS and GC-O. The concentrations of several odour- and flavour-active volatiles varied with ageing method and ageing time. The difference in flavour liking of aged beef could be explained by their corresponding volatile profiles. The preferred flavour liking of the dry aged beef samples described in a previous study could be attributed to some differences in volatile profiles, especially alkyl-pyrazines, 2-acetyl-2-thiazoline, and 3-hydroxy-2-butanone. However, the increased lipid oxidation and loss of desirable volatiles, e.g., 2-acetyl-1-pyrroline and alkyl-pyrazines, in the 56-day ageing can lead to the decrease in flavour liking. In wet aged beef, the accumulation of bacterial fermentation products appeared to be detrimental to its flavour. Therefore, based on the conditions used in this study, 35-day dry ageing is an optimal aging period to produce beef with a desirable volatile profile and flavour. Prolonged (56-day) ageing should be avoided in both dry ageing and wet ageing to prevent the deterioration of flavour, however further studies should be performed to confirm this.

## Figures and Tables

**Figure 1 foods-10-03113-f001:**
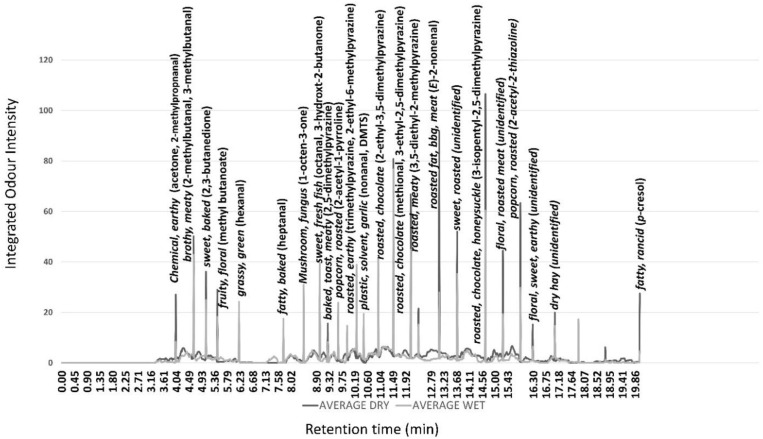
Average (*n* = 6 assessors) gas chromatography olfactometry profile of freshly grilled dry (black) and wet aged (grey) beef after 35 days.

**Figure 2 foods-10-03113-f002:**
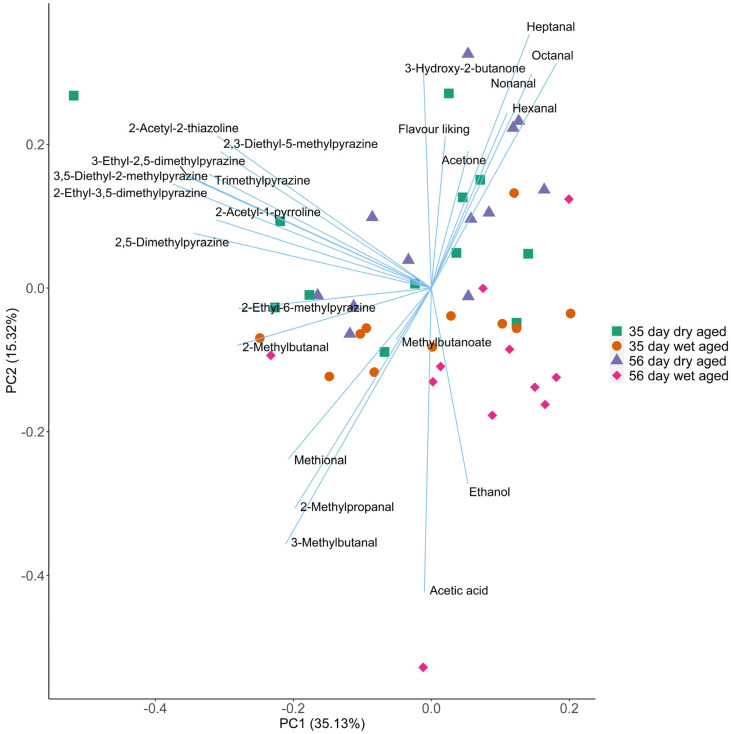
Principal component analysis (PCA) biplot showing volatile changes in beef dry aged for 35 days or 56 days, or wet 35 days or 56 days. Values on the axes refer to the variance explained by the first two principal components.

**Table 1 foods-10-03113-t001:** Effects of ageing method (AM; Dry aged, Wet aged) and ageing time (AT; 35 days, 56 days) on semi-quantitative data for odour-active volatiles concentrations (µg/kg) identified in the grilled aged beef headspace analysed with headspace solid phase microextraction-gas chromatography mass spectrometry. Values are means after adjustment for intramuscular fat (covariate). 4-Methyl-1-pentanol was used as an internal standard for semi-quantification purposes.

Volatile Compound	LRI	ID	Odour Descriptors	Lit	*m/z*	Dry Aged		Wet Aged		SED		*p* Value		
						35	56	35	56	AM/AT	AM × AT	AM	AT	AM × AT
**Alcohols/diols**														
Ethanol		EI,R		12	45	37	55	274	723	84.0	118.7	<0.001	0.007	0.013
1-Pentanol	945	EI,R		12	55	37.8	58.0	44.6	43.3	6.23	8.82	0.529	0.136	0.091
1-hexanol	1354	EI,R		12	56	69	105	59	19	31.3	44.3	0.133	0.956	0.228
1-Heptanol	1450	EI,R		12	70	21.2	43.4	20.5	12.5	10.98	15.53	0.156	0.525	0.177
2-Ethylhexanol	1496	EI,R		12	57	20.4	17.7	27.8	24.2	3.11	4.39	0.030	0.315	0.885
1-Octanol	1546	EI,R		12	56	9.3	17.2	10.4	8.0	2.87	4.06	0.162	0.352	0.080
4-Butoxybutanol	1701	EI,R			57	3.93	3.44	3.79	4.66	0.69	0.98	0.444	0.785	0.334
Benzylalcohol	1900	EI,R			108	13.3	32.3	4.3	6.3	4.06	5.74	<0.001	0.012	0.041
1-Ethylphenol	2044	EI,R			107	0.85	0.66	0.92	2.56	0.94	1.33	0.299	0.448	0.337
2-Ethylphenol	2194	EI,R			107	0.51	0.34	0.45	1.59	0.66	0.93	0.364	0.465	0.322
**Ketones/diones**														
Acetone	816	EI,R,O	Chemical, earthy		43	29.6	59.7	28.5	48.8	8.09	11.45	0.144	0.015	0.230
2-Butanone	886	EI,R		12	72	83.1	76.4	76.3	59.7	6.95	9.83	0.095	0.100	0.478
2-Pentanone	975	EI,R			86	44.1	49.2	63.4	69.5	7.54	10.67	0.011	0.462	0.948
2-Heptanone	1151	EI,R		8,15,30	58	6.76	4.98	3.31	9.67	1.38	1.96	0.654	0.104	0.005
2-Octanone	1238	EI,R		12	58	3.77	5.63	2.54	6.31	1.31	1.85	0.834	0.036	0.470
3-Hydroxy-2-butanone	1304	EI,R,O	Sweet, fresh, fishy	12,26	45	203	183	126	143	49.8	70.5	0.033	0.305	0.530
2-Methyl-3-octanone	1322	EI,R			99	3.77	5.63	2.54	6.31	1.31	1.85	0.834	0.036	0.470
2-Nonanone	1388	EI,R		8,15,30	58	2.71	6.18	3.31	3.83	0.79	1.12	0.275	0.015	0.067
1,3-Butanediol	1600	EI,R			45	1.5	1.6	3.4	6.9	2.02	2.86	<0.001	0.095	0.101
Butyrolactone	1637	EI,R			86	24.1	23.0	27.3	32.7	8.10	11.45	0.432	0.794	0.688
**Pyrazines**														
2-Methylpyrazine	1285	EI,R		8,12,15,30	94	27.6	25.4	19.6	17.4	4.47	6.32	0.081	0.624	0.999
2,5-Dimethylpyrazine	1330	EI,R,O	Baked, toast, meaty		108	100.2	82.7	78.4	55.5	14.78	21.91	0.103	0.177	0.857
2,6-Dimethylpyrazine	1338	EI,R		8,12,15,30	108	51.4	28.7	38.6	16.5	7.85	11.10	0.117	0.006	0.972
2,3-Dimethylpyrazine	1346	EI,R		8,12,15,30	108	11.28	8.07	5.66	5.07	1.89	2.67	0.026	0.319	0.490
2-Ethyl-5-methylpyrazine	1384	EI,R	Roasted, chocolate	12	121	10.73	7.23	6.66	4.32	1.02	1.44	0.001	0.006	0.571
2-Ethyl-6-methylpyrazine	1390	EI,R,O	Roasted, earthy	12	121	19.63	14.70	12.54	8.13	1.73	2.44	<0.001	0.009	0.880
Trimethyl pyrazine	1410	EI,R,O	Roasted, earthy	8,12,15,30	122	66.2	57.6	35.7	43.0	11.11	15.71	0.048	0.955	0.477
3-Ethyl-2,5- dimethylpyrazine	1442	EI,R,O	Roasted, chocolate	8,12,15,30	135	23.9	13.2	11.4	9.9	3.06	4.33	0.013	0.051	0.143
2-Ethyl-3,5-dimethylpyrazine	1469	EI,R,O	Roasted, chocolate	8,12,15,30	135	61.5	39.0	46.1	31.9	8.45	11.94	0.010	0.015	0.404
3,5-Diethyl-2- methylpyrazine	1490	EI,R,O	Roasted, meaty		149	5.95	3.50	3.04	2.30	0.80	1.13	0.013	0.051	0.291
2,3-Diethyl-5- methylpyrazine	1499	EI,R			149	3.01	0.98	0.57	0.72	0.52	0.73	0.012	0.075	0.040
3,5-Dimethyl-2-isobutylpyrazine	1549	EI,R			122	2.69	2.81	1.94	1.56	0.54	0.76	0.068	0.808	0.642
Dimethyl isopentylpyrazine	1655	EI,R		8,12,15,30	122	19.3	23.1	20.7	15.3	3.83	5.42	0.404	0.833	0.233
**Aldehydes**														
2-Methylpropanal	804	EI,R,O	Chemical, earthy	8,15,30	72	34.9	41.6	41.9	26.4	5.15	7.28	0.427	0.393	0.035
2-Methylbutanal	915	EI,R,O	Brothy, meaty	26	57	328	344	332	264	45.2	64.0	0.402	0.571	0.355
3-Methylbutanal	919	EI,R,O	Brothy, meaty	8, 15,30	58	113.8	119.3	142.4	111.9	15.69	22.19	0.502	0.427	0.257
Hexanal	1078	EI,R,O	Grassy, green	8,15,26,30	56	111	231	138	163	49.4	69.8	0.677	0.149	0.335
Heptanal	1194	EI,R,O	Fatty, baked	8,15,26,30	70	20.3	78.2	34.7	41.2	12.66	17.90	0.375	0.014	0.047
Octanal	1328	EI,R,O	Sweet, fresh fish	8,15,26,30	84	8.0	18.9	14.2	9.4	2.50	3.53	0.517	0.222	0.003
Nonanal	1380	EI,R,O	Plastic, solvent, garlic	8,15,26,30	57	58.5	94.1	92.1	59.5	9.98	14.12	0.958	0.879	0.001
Furfural	1439	EI,R		8	96	1.9	2.1	1.8	6.1	2.15	3.04	0.376	0.311	0.343
Decanal	1500	EI,R		8,12,15,30	57	0.19	0.70	0.31	0.61	0.19	0.27	0.943	0.036	0.567
Benzaldehyde	1508	EI,R		8,12,15,30	105	57.1	52.9	71.2	81.3	8.33	11.78	0.014	0.725	0.397
2,5-Dimethylbenzaldehyde	1705	EI,R			134	177	70	210	169	27.2	38.5	0.019	0.009	0.224
4-Ethylbenzaldehyde	1732	EI,R			134	2.41	2.85	1.96	5.35	1.46	2.07	0.486	0.196	0.319
Long chain aldehyde	1736	EI,R			57	38	21	56	73	7.6	10.8	0.001	0.924	0.052
**Sulphur compounds**														
Dimethyl disulphide	1084	EI,R		8,12,15,30	94	21.2	4.1	8.7	8.4	4.54	6.42	0.374	0.059	0.069
Dimethyl trisulphide	1266	EI,R,O	Plastic, solvent, garlic	8,30	79	23.1	0.1	0.9	0	10.48	20.97	0.291	0.260	0.296
Methional	1447	EI,R,O	Roasted, chocolate	8,15,30	76	1.93	2.16	3.66	2.55	0.63	0.89	0.099	0.484	0.289
Methionol	1717	EI,R			106	0.83	0.53	1.17	0.82	0.28	0.40	0.263	0.260	0.934
2-Acetyl-2-thiazoline	1756	EI,R,O	Popcorn, roasted	8,15,30	129	1.31	1.14	0.75	0.14	0.22	0.31	0.001	0.091	0.333
Benzothiazole	1955	EI,R			135	43	32	38	91	32.0	45.2	0.411	0.512	0.319
**Acids**														
Acetic acid	1461	EI,R		12	60	38	34	104	148	24	34	<0.001	0.412	0.317
**Esters**														
Methyl butanoate	978	EI,R,O	Fruity, floral	8,15,30	74	76.8	58.5	78.5	64.6	7.43	10.50	0.603	0.035	0.769
Butyl formate	996	EI,R			56	147.0	109.8	148.4	112.0	11.99	16.96	0.881	0.003	0.972
Methyl-2-methylbutanoate	1008	EI,R			88	20.37	14.14	19.45	16.07	1.92	2.72	0.795	0.016	0.461
Ethyl nonanoate	1520	EI,R			74	4.53	6.62	5.67	5.53	1.04	1.47	0.981	0.354	0.291
Methyl salicylate	1747	EI,R			120	4.7	4.7	4.2	17.1	6.55	9.26	0.368	0.328	0.328
**Others**														
Pyridine	1204	EI,R			79	31.7	31.3	24.6	18.0	2.81	3.98	<0.001	0.214	0.270
2-Pentylfuran	1250	EI,R		8,15,30	81	2.93	10.32	3.06	7.98	1.54	2.17	0.474	<0.001	0.425
Pyrrole	1524	EI,R			67	5.04	3.68	3.97	5.18	0.74	1.05	0.965	0.727	0.0052
2-Acetyl-1-pyrroline	1998	EI,R,O	Popcorn, roated	8,15,30	94	8.55	5.58	7.93	4.20	0.78	1.11	0.205	<0.001	0.628

SED: standard error of difference; *m/z* = mass to charge ratio of ion used for quantification; ID = method of identification; EI = positive match of the electron impact mass spectrum in the NIST mass spectral library; R = retention index match with reference; O = odour quality by gas chromatography-olfactometry; LRI = linear retention index; Lit = reference number for the literature in which the volatiles were identified.

**Table 2 foods-10-03113-t002:** Estimated regression coefficients from partial least squares models for prediction of flavour liking using concentrations of selected odour-active volatiles in aged beef. Models based on one latent variable or factor.

Variables	Regression Coefficient
Constant	69.9888
2-Ethyl-6-methylpyrazine	0.3508
2,5-Dimethylpyrazine	0.3466
2-Ethyl-3,5-dimethylpyrazine	0.344
2-Acetyl-1-pyrroline	0.3343
3,5-Diethyl-2-methylpyrazine	0.3189
3-Ethyl-2,5-Dimethylpyrazine	0.317
2-Acetyl-2-thiazoline	0.2976
2-Methylbutanal	0.2880
2,3-Diethyl-5-methylpyrazine	0.2820
Butylformate	0.2718
Ethanol	−0.2631
Acetic acid	−0.2569
3-Hydroxy-2-butanone	0.2556
1-Hexanol	0.2547
Methylbutanoate	0.2314
2-Methylpropanal	0.1713
Acetone	−0.1299
Dimethyl trisulfide	0.1238
Heptanal	−0.0908
Nonanal	0.0764
Octanal	−0.045
Hexanal	0.0442
s.d.	4.61
Osten’s *F*-test	<0.001
RMSECV	2.31
Correlation coefficient	0.94

s.d.: standard deviation. RMSECV: root mean square error of cross validation.

## Data Availability

The data presented in this study are available on request from the corresponding author. The data are not publicly available at this moment because they are part of an ongoing Ph.D. Thesis.

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
