# Peer review of "Volatile Profile of Dry and Wet Aged Beef Loin and Its Relationship with Consumer Flavour Liking"

_foods, 2021, doi:10.3390/foods10123113_

Round 1

Reviewer 1 Report

Summary: The authors investigating the effect of differing ageing times and methods on the volatile profile of beef striploin steaks using SPME-GMCS. GC-O was used to identify particularly important odour compounds, something that elevates the quality of this research above other volatile profile research just employing SPME-GC methods. Overall, the research is well-thought and carried out. There are just minor corrections/additions to the Methods & Materials and the Results and Discussion sections necessary. 

Specific comments. 

Line 70: I understand that M&M information may bein included in Ha et al.; however please include information to the physiology of the carcasses, such as breed, age, sex, etc. 

Line 81: Please state whether cuts were aged with or without illumination. If with illumination, please include information on light source and length of time per day.  

Line 91: please state for how long samples were stored at -20C prior to GC and sensory analysis. 

Line 103: leads to confusion/conflicts with/and repeates the information in line 80. Please delete the sentence at line 103 and insert the information ”randomly selected“ to line 80. 

Line 199: please rephrase and replace “process“ with “time“ to keep with chosen terminology used within the text. Or, alternatively, phrase as “during the ageing process.” The current word choice is confusing and ambiguous.    

Line 230 -237: the relation with TBARS values and the “possible” explanations are not sufficiently clearly explained. Especially the context of the last sentence is missing; is this in reference to the dry-aged or wet-aged TBARS values? Instead of reaching for possible explanations, it would be preferable to objectively state the observation and explain that the relationship between volatile compounds and TBARS is not clear. 

Line 246: delete the comma in front of “were”

Line 259: please provide at source for the comment “increased the oxidation”, does this statement originate from the TBARS values measured in Ha et al.? If there is no source for this statement, please delete the postulation. 

Line 299: “M. Biceps femoris” should be italicized

Line 301: “longissimus thoracis et lumborum” should be italicized

Lin 330: please replace “2017” with “(15)” the corresponding citation style

Sub-Sections in Results and Discussion: it would be easier for the reader, if the section was divided into smaller paragraphs per result and associated discussion.    

Author Response

The authors would like to thank the Reviewer for your time to help improving the manuscript. Please see our response below.

Summary: The authors investigating the effect of differing ageing times and methods on the volatile profile of beef striploin steaks using SPME-GMCS. GC-O was used to identify particularly important odour compounds, something that elevates the quality of this research above other volatile profile research just employing SPME-GC methods. Overall, the research is well-thought and carried out. There are just minor corrections/additions to the Methods & Materials and the Results and Discussion sections necessary. 

We appreciate your time and effort.

Specific comments. 

Line 70: I understand that M&M information may be included in Ha et al.; however please include information to the physiology of the carcasses, such as breed, age, sex, etc. 

We have added information on the carcasses in line 71-73.

Line 81: Please state whether cuts were aged with or without illumination. If with illumination, please include information on light source and length of time per day.  

Thanks, we have now added this as follows. ‘The wet ageing was conducted without illumination’ (line 86). The dry ageing was conducted with UV lights, which is stated in the original manuscript in line 88.

Line 91: please state for how long samples were stored at -20C prior to GC and sensory analysis. 

The GC samples were stored at -80 °C for 4 months and the sensory samples were stored at -20°C for 3 months.  This data has now been added in the text in line 95-96.

Line 103: leads to confusion/conflicts with/and repeates the information in line 80. Please delete the sentence at line 103 and insert the information” randomly selected“to line 80. 

We agree, we have deleted the sentence from line 107 and inserted information at line 84.

Line 199: please rephrase and replace “process“ with “time“ to keep with chosen terminology used within the text. Or, alternatively, phrase as “during the ageing process.” The current word choice is confusing and ambiguous.    

We agree, we have replaced it with “time” in line 209.

Line 230 -237: the relation with TBARS values and the “possible” explanations are not sufficiently clearly explained. Especially the context of the last sentence is missing; is this in reference to the dry-aged or wet-aged TBARS values?

This is in reference to the wet-aged TBARS value, we have now added this as follows. ‘Whereas the malondialdehyde measured by TBARS is unlikely to be affected by the microorganisms in meat during the wet ageing.’ (line 249).

Instead of reaching for possible explanations, it would be preferable to objectively state the observation and explain that the relationship between volatile compounds and TBARS is not clear. 

Thanks. We have now added “Therefore, the relationship between volatile compounds and TBARS was not clear in the present study” in line 244-245, but 

Line 246: delete the comma in front of “were”

Thanks, we have deleted it in line 260.

Line 259: please provide at source for the comment “increased the oxidation”, does this statement originate from the TBARS values measured in Ha et al.?

Thanks, and yes, this statement originated from the TBARS values measured in Ha, et al. 2019. We have added this to the manuscript, line 274.

If there is no source for this statement, please delete the postulation.

We have now added “, as Ha, et al. [3] showed that the dry aged beef had significantly higher TBARS than the wet aged beef at day 56 (p < 0.05)” in line 274-275.

Line 299: “M. Biceps femoris” should be italicized

Thanks, we have changed it accordingly in line 316.

Line 301: “longissimus thoracis et lumborum” should be italicized

Thanks, we have changed it accordingly in line 318.

Lin 330: please replace “2017” with “(15)” the corresponding citation style

Thanks, we have changed it accordingly in line 348.

Sub-Sections in Results and Discussion: it would be easier for the reader, if the section was divided into smaller paragraphs per result and associated discussion.    

We agree, we have now divided the section “3.1 Grilled beef volatile profiles” into three paragraphs.

Reviewer 2 Report

This is a very nice piece of work. However, I have some comments, questions and suggestions detailed below:

The MANOVA model used should be specified and detailed.

The PLS model should be performed using standardized data (please use z-scores).

The PCA is not clear. Can the authors add the individuals instead of one point for each group. The 24 samples per group should be shown here? Otherwise, specify the number of samples per group that is not very clear in the manuscript.

Is the SED of Table 1 necessary ?

What is the overall MSA of Figure 2? Are all the variables eligible to be maintained in the PCA? Please check.

Are all the variables in the model of Table 2 eligible to enter the model? In other words, are all those variables explanatory? I suspect overfitting. What about the principle of parsimony? Please use a multiple regression model in a stepwise manner to retain the eligible variables only? I would suggest you to present the Table in a graph to easily see the variables that are negative and positive. Also rank them based on their power.

Can the authors specify how they handled multicollinearity in this study? Nothing was mentioned, and this is questionable, especially in Table 2.

Author Response

This is a very nice piece of work.

Thank you. We appreciate your time and effort in helping to improve the manuscript.

However, I have some comments, questions and suggestions detailed below:

The MANOVA model used should be specified and detailed

Thanks, the MANOVA model used in our study was as follow. Response = ageing method + ageing time + ageing method * ageing time + animal (block) + intramuscular fat content (covariate). This had been stated previous in the original manuscript as “The multivariate analysis of variance (MANOVA) procedure was performed on the volatile data using ageing method, ageing time as main effects, and the intramuscular fat content of each carcass was used as covariate, and the animal was used as the block.” in line 161-165.  

The PLS model should be performed using standardized data (please use z-scores).

Thanks, we did use the z-scores ((value-mean/standard deviation)) in the present PLS model. We’ve added the information to line 175-176 to make it clearer.

The PCA is not clear. Can the authors add the individuals instead of one point for each group. The 24 samples per group should be shown here? Otherwise, specify the number of samples per group that is not very clear in the manuscript

Thanks, but we do not agree. We have run the PCA using individuals, but the biplot was not very clear due to too many data points and loadings. So, we used the mean data for each treatment group instead to make it easier for the readers to look at the trend and follow the logic of the discussion. We have added this information to the method part in line 169-172.

Is the SED of Table 1 necessary?

We think it is appropriate to keep the SED.  It is good statistical methodology approach to give the SED’s for the main effects and interaction.

What is the overall MSA of Figure 2?

We are not sure what you mean by MSA. But if what you mean is the MQ4 score, it is affected by other sensory quality such as tenderness and juiciness, which are irrelevant to this paper. This paper only used the MSA flavour liking scores.

 Are all the variables eligible to be maintained in the PCA? Please check.

Thanks, we have tried to maintain all the variables eligible, but the PCA biplot was crowded with the data (observation) points, loading plot, and loading labels, which made the figure very difficult to read. So, we decided to only use volatiles with odour impacts in GC-O as mentioned in the method in line 169-172.

Are all the variables in the model of Table 2 eligible to enter the model? In other words, are all those variables explanatory?

Thanks, we believe all the variables in the model are explanatory. We established this PLS model following the method reported by Frank, et al., 2017 (please see the reference below). Firstly, we put the all the volatiles into the PLS model to check their loadings on the PLS components, and we used two approaches in the selection of the variables. 1. The PLS component loadings were used to select the most influential volatiles. 2. Among these influential volatiles, we selected sub-set of volatiles known to be odour-active in our GC-O results or results from literatures for the final model. We’ve now added this information and rephrased the statistical analysis part to avoid confusion in line 172-181. Also, we have now added the GC-O information and literature reference of these volatiles to Table 1.

Frank, D.; Raeside, M.; Behrendt, R.; Krishnamurthy, R.; Piyasiri, U.; Rose, G.; Watkins, P.; Warner, R. An integrated sensory, consumer and olfactometry study evaluating the effects of rearing system and diet on flavour characteristics of australian lamb. Animal Production Science 2017, 57, 347-362.

I suspect overfitting.

We believe the data was not overfitted for the following reasons; 1. The input variables into the PLS model were carefully selected as mentioned above. 2. We only used one PLS component (line 181), and we understand that if we used more PLS components, the overfitting might be a problem. 3. The Osten’s F-test tests for this, and the result showed that the data was not overfitted as shown in Figure 2.

What about the principle of parsimony?

Thanks, and yes, we did ensure parsimony. As mentioned above, we have a strict standard in the selection of input variables into the model.

Please use a multiple regression model in a stepwise manner to retain the eligible variables only?

Thanks, but PLS is an effective and widely used methods to study the relationship between volatiles and flavour of food. Also, there are too many variables (62 volatiles) for a stepwise multiple regression.

I would suggest you to present the Table in a graph to easily see the variables that are negative and positive.

Thanks, but as the coefficients are shown with their positive/negative signs, we do not see a need to include a graph. We also think a table is more informative.

Also rank them based on their power.

Thanks for your suggestion. We have now done this in Table 2.

Can the authors specify how they handled multicollinearity in this study? Nothing was mentioned, and this is questionable, especially in Table 2.

Multivariate regression including partial least-square regression and principal component regression are commonly used methods to overcome the multicollinearity, as the linear combination of variables instead of themselves are used in these regress models, in our case, it was the PLS components. There is no collinearity between each PLS component as all PLS components are orthogonal, and we only used one PLS component in our model.

Reviewer 3 Report

The topic of manuscript entitled: "Volatile profile of dry and wet aged beef loin and its relationship with consumer flavour liking" could interest readers however it has a significant flaws. The authors seems not understand the terms of volatolomics and sensory analysis. The research is not conducted with standard procedure for olfactometry. 

For example:

Abstract include results with description "decreased" or "increased" but he question is how much?

"sensory flavour liking"? what means that phrase? It is out of nomenclature.

What means "The concentration many flavour impact volatiles"?

The Authors don't present "aromagrams" in Figure , this is just chromatogram with assigned aroma characteristics. 

Table 1. "maillard" it is a name and this a big shortcut. "Maillard reaction products" or "Maillard-derived volatiles".

Below Table 1. "EI- electron ionization" where the Authors used such information.

It is written"more than 60 volatiles", there is exactly 62 compounds.

Proper is to write HS-SPME GC/MS which means headspace solidphase microextraction gas chromatography with mass spectrometry, and this is wrongly written in several places in the text.

Authors wrote "sensory" and "consumer" but there was no sensory or consumer study.

Why 4-methylpentanol was used as internal standard? Did authors establish parameters of HS-SPME by themselves or based of another studies?

The SPME always has a two step: incubation and extraction, what are parameters of incubation in this study?

How long was SPME fiber hold in GC port?

"GC-MS-O" it is not appropriate nomenclature "GC-O/MS"

In the Table 1 the volatiles are grouped according to molecule classes, what for?

There is no results for GC-O. Analysis are done only using one (polar) column. The standard procedure of aroma characterization requires using of two columns: polar and non-polar. No Factor dilutions (FD), odor thresholds (O) and odor activity values (OAVs). In the keywords it is written "odour active"? The Authors has a serious problems with aroma analysis nomenclature. 

Author Response

The topic of manuscript entitled: "Volatile profile of dry and wet aged beef loin and its relationship with consumer flavour liking" could interest readers however it has a significant flaw. The authors seems not understand the terms of volatolomics and sensory analysis. The research is not conducted with standard procedure for olfactometry. 

Thank you for your time reviewing the paper. Co-authors Frank and Warner have many papers published in sensory, volatile analysis and olfactory using similar techniques. We believe our olfactometry as well as the other aroma analytical methods, are reliable and valid.

For example:

Abstract include results with description "decreased" or "increased" but he questions is how much?

Thanks, we have now added the significance and p value to the abstract.

"sensory flavour liking"? what means that phrase? It is out of nomenclature.

Thanks, we have now reworded all “sensory flavour liking” to “consumer flavour liking” in the manuscript, which is a nomenclature in the Meat Standard Australia (MSA) sensory protocol which has been adopted by UNECE as an international standardised consumer sensory testing protocol for meat. Our study used the MSA sensory protocol with the sensory results published in Ha et al., 2019 (reference [3] of our manuscript).

What means "The concentration many flavour impact volatiles"?

Thank you for pointing this out. We have now reworded all “flavour impact volatiles” to “odour impact volatiles” in the manuscript.

The Authors don't present "aromagrams" in Figure 1, this is just chromatogram with assigned aroma characteristics. 

The authors disagree as this figure actually is an aromagram. The data in the figure are based on average perceived odour intensity at the sniff port (n=6 assessors). However, to clarify this point, we have now added the “n=6 assessors to line 302” and we have also added labels to axes. We apologise for the oversight. 

Table 1. "maillard" it is a name and this a big shortcut. "Maillard reaction products" or "Maillard-derived volatiles".

Thanks, we have now deleted the “Maillard” but kept the “Pyrazines” in Table 1.

Below Table 1. "EI- electron ionization" where the Authors used such information.

Thanks, “EI means the “positive match of the electron impact mass spectrum in the NIST spectral library. “We have now added relevant information to Table 1 (line 254).

It is written"more than 60 volatiles", there is exactly 62 compounds.

We have changed this sentence in line 188 to "A total of 62 volatiles..."

Proper is to write HS-SPME GC/MS which means headspace solidphase microextraction gas chromatography with mass spectrometry, and this is wrongly written in several places in the text.

Thanks, we have changed them to HS-SPME GC-MS, similar to nomenclature in previous studies of Warner and Frank, in the whole manuscript.

Authors wrote "sensory" and "consumer" but there was no sensory or consumer study.

As we stated in the method part (line 96-98), the consumer flavour liking data was obtained from the study of Ha, et al., 2019 (reference [3] in the manuscript).

Ha, M.; McGilchrist, P.; Polkinghorne, R.; Huynh, L.; Galletly, J.; Kobayashi, K.; Nishimura, T.; Bonney, S.; Kelman, K.R.; Warner, R.D. Effects of different ageing methods on colour, yield, oxidation and sensory qualities of australian beef loins consumed in australia and japan. Food Research International 2019, 125, 108528.

Why 4-methylpentanol was used as internal standard? Did authors establish parameters of HS-SPME by themselves or based of another studies?

This internal standard has been used in previous studies, including those published by co-authors Warner and Frank. Please see a couple of references below. We have now clarified it as 4-methyl-1-pentanol in the method (line 113) to make it clearer.

Frank, D.; Ball, A.; Hughes, J.; Krishnamurthy, R.; Piyasiri, U.; Stark, J.; Watkins, P.; Warner, R. Sensory and flavor chemistry characteristics of australian beef: Influence of intramuscular fat, feed, and breed. Journal of Agricultural and Food Chemistry 2016, 64, 4299-4311.

Frank, D.; Raeside, M.; Behrendt, R.; Krishnamurthy, R.; Piyasiri, U.; Rose, G.; Watkins, P.; Warner, R. An integrated sensory, consumer and olfactometry study evaluating the effects of rearing system and diet on flavour characteristics of australian lamb. Animal Production Science 2017, 57, 347-362.

The SPME always has a twostep: incubation and extraction, what are parameters of incubation in this study?

Thank you for pointing this out. The samples were pre-incubated at 40°C for 15 min and then extracted at 40°C for 40 minutes. We have added this information to the method part in line 115.

How long was SPME fiber hold in GC port?

The SPME fibre was desorbed in GC port for 5 minutes, we have now added this information to the method part in line 119.

"GC-MS-O" it is not appropriate nomenclature "GC-O/MS"

There are many variations of this nomenclature in the literature. However, However, we have changed it to GC-O-MS to keep it consistent with our previous published studies, along with many others.

In the Table 1 the volatiles are grouped according to molecule classes, what for?

The volatiles are grouped according to molecule classes to help the readers to follow the logical flow of discussion. And now we have changed the order of volatiles in each group based on their LRI (from low to high) in Table 1 to make it clearer.

There is no results for GC-O. Analysis are done only using one (polar) column. The standard procedure of aroma characterization requires using of two columns: polar and non-polarIn the keywords it is written "odour active"?

We do not agree. There are many prior published studies where only a WAX column was used. We used LRI and EI ionisation mass spectra to confirm IDs in most cases as well as odour character. We also checked against published LRIs for wax columns (NIST Chemistry Webbook). We have now added this information to the method in lines 134-135 and 151-154 to make it clearer. Co-authors Warner and Frank have published many studies in peer-reviewed journals using these methodologies.

No Factor dilutions (FD), odor thresholds (O) and odor activity values (OAVs).

Direct intensity olfactometry methods are widely used and is an acceptable approach. Flavour dilution analysis is prohibitively time consuming and not widely practised. The odour-active volatiles identified in this study have been widely reported by us and others in grilled meat previously.

Please see Delahunty, C. M., Eyres, G., & Dufour, J.-P. (2006). Gas chromatography-olfactometry. Journal of Separation Science, 29(14), 2107-2125. doi: https://doi.org/10.1002/jssc.200500509.

The Authors has a serious problems with aroma analysis nomenclature. 

We disagree. Our co-authors Frank and Warner have extensive expertise in aroma analyses and meat flavour chemistry, evidenced by many published studies in this research area. We believe most of the points on nomenclatures the Reviewer raised were simply variations in naming of the techniques.

Round 2

Reviewer 2 Report

The reviewer understands that the MANOVA model was based on previously papers, but it is mandatory to clarify it in this paper. The paper should be clear without need to access other papers. Please add the full model.

The graphs of Figure 1 are of very low quality. Please move to the supplementary data.

The reviewer don't agree to see only the barycentres of the groups in the PCA bi-plots. Please add the individuals or a minimum the circle of the variations for each of them. The authors can present the PCA results (two first axes) separately of the bi-plot. It is important to see the extent of the separation.

Based on the PCA analysis, we can see that some variables are very close to each other, hence suspecting a multicollinearity. The authors should add in the supplementary file the results of the multicollinearity test of the regression model that would confirm if there is or no overfitting. Also, check the suitability of keeping all the variables in the PCA based on MSA test (Measure of Sampling Adequacy: MSA).

Are all the variables of the model in Table 2 significant in the model? Please add a column with the p-values of each variable. Also, this model should consider the parsimony principle where a number of variables should be limited. A step-wise regression model is more suitable in this study.

Author Response

The authors would like to thank the Reviewer for your effort and constructive comments. We have previously checked our statistical approach with an expert statistician (Prof. Graham Hepworth, https://scc.ms.unimelb.edu.au/people/graham), who has over 30 years of consulting experience on similar studies), who has agreed that our approach is appropriate and statistically sound. Hence, we are confident of our statistical approach.

Q: The reviewer understands that the MANOVA model was based on previously papers, but it is mandatory to clarify it in this paper. The paper should be clear without need to access other papers. Please add the full model.

A: Thanks, we have reworded the description of the MANOVA method to “The multivariate analysis of variance (MANOVA) was performed on the volatile data using the following model, ” Response (volatile data) = ageing method (main effect) + ageing time (main effect) + ageing method * ageing time (interaction) + animal (block) + intramuscular fat content (covariate).” “ in line 161-166 to clarify this.

Q: The graphs of Figure 1 are of very low quality. Please move to the supplementary data.

A: Thanks, we have now reformatted Figure 1 to high resolution (1000 dpi) so it can be kept in the main part of the paper.

Q: The reviewer don't agree to see only the barycentres of the groups in the PCA bi-plots. Please add the individuals or a minimum the circle of the variations for each of them. The authors can present the PCA results (two first axes) separately of the bi-plot. It is important to see the extent of the separation.

A: Thanks, we have now used a PCA biplot fitted with individual data points. And 12 sample per group is shown in the graph as described in line 83-84 except for one missing data point in “56-day dry aged” and one in “56-day wet aged” groups respectively due to missing sensory data. We have added relevant information to line 172 -174 to make it clearer. We have also made corresponding changes to the result and discussion part in line 318-321,322, 326, 335.

Q: Based on the PCA analysis, we can see that some variables are very close to each other, hence suspecting a multicollinearity.

A: Thanks, the authors agree that some variables share multicollinearity, for example, the pyrazines are usually derived from the Maillard reaction and the Strecker degradation, and the hexanal, heptanal, and octanal are derived from the oxidation of unsaturated fatty acids in meat. So, it is natural and normal for these compounds to have multicollinearity, and this is a very common issue in flavour chemistry studies. And that’s why partial least-square regression is commonly used in the literature and in flavour chemistry/sensory studies to overcome this issue. Hence this is why we used it in the present study. Please see the explanation in detail in the following section below.

Q: The authors should add in the supplementary file the results of the multicollinearity test of the regression model that would confirm if there is or no overfitting.

A: Thanks, but the authors don’t think a multicollinearity test is needed for the partial least-square regression (PLS). PLS is actually designed to handle data with multicollinearity. In the process of partial least-square regression, all the variables were firstly combined into a linear combination of them, which is known as the PLS component or the latent variable as follow:

Z = VX

Where Z is the latent factor matrix, and X is the matrix that contain all the variables, V is the loading matrix. This process is a bit similar to PCA, but the loading was calculated based on the covariance between the predictors (X) and the Reponses (Y) instead of the variance in X.

Then the latent factor was used as the predictor as follow:

Y = βzZ

Where βz is the regression coefficient of the latent factor on the Y (response).

All the latent factors (PLS1, PLS2, PLS3,,,) used in PLS are orthogonal, so there is no need to test the multicollinearity of them. And the multicollinearity between volatile variables in this study was overcome by combing them into latent factor as above.

For this reason, the PLS is widely used to overcome the multicollinearity in variables. Please see the following examples:

Carrascal, L. M., Galván, I., & Gordo, O. (2009). Partial least squares regression as an alternative to current regression methods used in ecology. [https://doi.org/10.1111/j.1600-0706.2008.16881.x]. Oikos, 118(5), 681-690. doi: https://doi.org/10.1111/j.1600-0706.2008.16881.x

Farahani, H. A., Rahiminezhad, A., Same, L., & immannezhad, K. (2010). A Comparison of Partial Least Squares (PLS) and Ordinary Least Squares (OLS) regressions in predicting of couples mental health based on their communicational patterns. Procedia - Social and Behavioral Sciences, 5, 1459-1463. doi: https://doi.org/10.1016/j.sbspro.2010.07.308

And the mechanism and principles for partial least-square regression:

Hubert, M., & Branden, K. V. (2003). Robust methods for partial least squares regression. [https://doi.org/10.1002/cem.822]. Journal of Chemometrics, 17(10), 537-549. doi: https://doi.org/10.1002/cem.822

Therefore, the multicollinearity between the original variable will not cause overfitting. But we understand that the use of  more PLS components (PLS2,PLS3,,,) can also cause overfitting, and that’s why we only used one PLS component in the model as described in line 188-189.

Q: Also, check the suitability of keeping all the variables in the PCA based on MSA test (Measure of Sampling Adequacy: MSA).

A: Thanks for your advice on the Measure of Sample Adequacy. We have checked the suitability of keeping all the variables in the PCA based on MSA test (Kaiser-Meyer-Olkin test). When we put the selected variables described in line 172-174, the KMO was 0.67, which indicates that the variables are suitable for PCA , and we have added this result to line 318-319. And if all the variables are kept in the PCA, the KMO was less than 0.41, which indicate that it is inappropriate for factor analysis or PCA. Therefore, the results of the MSA further confirms that the selection of variable in our PCA is valid and not all the variables should be kept. We appreciate your advice and have added relevant information to line 177-179.

Q: Are all the variables of the model in Table 2 significant in the model? Please add a column with the p-values of each variable.

A: Thanks, but conventional significance test is not feasible in PLS for the following reasons. 1. As we mentioned in the answer for an earlier comment, the variable used in the regression model is actually the PLS component, and the distribution of it is unknown. 2. Even the p-value of the PLS in the model can be calculated using some special statistical test (usually involve resampling and can cause high bias), the p-value for each original variable (volatiles in our case) still cannot be calculated. For these reasons, p-value is usually not calculated and presented in the PLS model. Please see relevant information in the following paper:

Pirouz, D. M. J. A. a. S. (2006). An overview of partial least squares.

Q: Also, this model should consider the parsimony principle where a number of variables should be limited.

A: Thanks, we did consider the parsimony principle by setting strict criteria in the selection of predictor variables as we mentioned in the method part in line 180-186. As a result, only 22 out of 62 volatiles were selected to enter the model. We have now added n =22 to line 185 to make it clearer.

Q: A step-wise regression model is more suitable in this study.

A: Thanks, and the authors appreciate your suggestion on the step-wise regression but hold a different opinion on that. We understand that stepwise regression is a valid and effective method in many cases, however, it’s not suitable for our study for the following two reasons:

1. Stepwise regression usually rely on the selection algorithms to exclude variables and retain a small number of predictor variables in the final model. We understand that you suggest us to use this method to meet the parsimony principle. However, the flavour/odour of cooked meat is usually influenced by a large group of volatiles rather than few volatiles. In other words, the difference in meat flavour can’t be attributed only to few volatiles. Therefore, it is preferable to include a reasonable number of volatiles in the model to explain the difference in flavour liking. And when a relatively large group of variables (especially volatiles) are used in the model, multicollinearity become a prominent issue, and that’s one of the reasons why we chose to use PLS as we mentioned above.

2. In the selection of variables in our model, we considered their odour impact based on results in our GC-O analysis or previous studies (line 180-186). This practice is a priori. In stepwise regression, the selection of variables was performed by firstly fitting in all the variables and then exclude or retain them based on the results of statistical analysis (eg, p-value), and this is sometimes considered as post hoc. For these reasons, we believe our variable selection criteria and model are more suitable in this study.

Reviewer 3 Report

I admit that the Authors make significant corrections in comparison to the previous version. However, please refer to some minor suggestions.

How many untrained consumer panelist took part in your study? what sensory attributes were selected?

Table 1 should contain the odour descriptor determined by GC-O.

I don't know if something changed but in Molecules "Results and Discussion" is a chapter before "Materials and methods".

Author Response

Q: I admit that the Authors make significant corrections in comparison to the previous version. However, please refer to some minor suggestions.

A: Thank you for your comments, we appreciate the efforts taken by the reviewer.

Q: How many untrained consumer panelist took part in your study?

A: 900 untrained consumer panelists took part in Ha, et al.’s study. We have now added “with 900 participants” to line 97-98.

Q: what sensory attributes were selected?

A: Four sensory attributes, tenderness, juiciness, flavour liking, and overall liking were selected based on the Meat Standard Australia (MSA) protocol. We didn’t include this information because only flavour liking was relevant to this paper, and the untrained consumer panel was described in detail in the study of Ha, et al.

Q: Table 1 should contain the odour descriptor determined by GC-O.

A: Thanks, we have now added a column “Odour descriptors” in Table 1.

Q: I don't know if something changed but in Molecules "Results and Discussion" is a chapter before "Materials and methods".

A: Thanks, but the “Materials and methods” section goes before the “Results and Discussion” section as a standard format required by the journal Foods.